# Evading Web Application Firewalls with Reinforcement Learning

**Xianbo Wang**
Department of Information Engineering
The Chinese University of Hong Kong
xianbo@ie.cuhk.edu.hk

**Han Hu**
Department of Information Engineering
The Chinese University of Hong Kong
hh118@ie.cuhk.edu.hk

## Abstract

Web Application Firewalls (WAF) are widely deployed to protect web servers from security threats like SQL injections. WAF products employ various techniques, e.g., syntax signature and machine learning, to detect and block suspicious web traffics. However, no WAF can be absolutely secure, there are always space for adversaries to craft malicious messages that can evade the detection. In the past, most evasion techniques are developed manually, which requires labour and intelligence. In this work, we propose to explore the possibility of automating the process of WAF evasion using reinforcement learning. We created a reinforcement learning environment (based on OpenAI gym) for WAF evasion tasks and evaluate various mainstream WAF products with Proximal Policy Optimization (PPO) algorithm. Our framework successfully discovered numbers of evasion payloads for each WAF in our experiments and can significantly outperform baseline policy. Finally, we extract common patterns from the discovered evasion payloads and discuss weaknesses/flaws of existing WAF products as well as suggested improvements. [1]

## 1 Introduction

Web application attacks are the single most prevalent and devastating security threat facing organizations today. By exploiting common vulnerabilities like SQL injection and Cross Site Scripting (XSS) in web applications, cybercriminals can steal private data or even gain full control of the server, which can cause huge financial loss. To mitigate the threats, many organizations deploy the Web Application Firewalls (WAF) to block suspicious web traffics and protect their web servers.

In general, current WAF products can be divided into two types, namely, rule based and machine learning based. Rule based WAFs check the HTTP messages with predefined syntax patterns, e.g., with regular expressions, while machine learning based WAFs extract various features from the traffic data and run classifiers to determine whether to block the traffic. For both methods, it is unrealistic to block all malicious traffics as there are infinitely many variations. Take SQL injection for example, based an ordinary attack payload `admin' OR 1=1#`, which can bypass the password checking query `SELECT * FROM users WHERE name='x' pwd='y'`, adversaries can craft variations like shown in Listing 1. While appearing differently, these variations all share the exact same semantic with the original payload.

```
1  Original: admin' or 1=1#
2  Variant1: admin'oR(sEleCt 1)=0x1-- -
3  Variant2: admin'/**/oR/**/(/*!%53eLEct*/1)>0x0#
```
Listing 1: Variants of a SQL Injection Payload

---

[1]Our 5-minutes video presentation can be accessed at https://bit.ly/3gGwfBa with password: DE@DB33F

As most commercial WAF products do not publish their model or algorithm, for generality, we assume adversaries have no knowledge of the internal states of the WAF. To craft a payload variation that can evade the detection of an WAF, the adversary needs to send each mutation to the WAF, observe its reaction, for which we consider two different scenarios:

- Black-box: the adversary can only observe the Boolean result, that is payload either being detected or not by the WAF.
- Grey-box: apart from the Boolean result, the adversary can also get a scalar score, e.g., a confidence level, of the classification result in the WAF.

Then, based on the observation, the adversary keeps mutating the payload until it can evade the WAF's detection. While this process of payload manipulating and testing is repetitive and painful for human, it seems suitable for reinforcement learning, simply by mapping payload mutation to agent's action and mapping WAF's reactions as rewards.

In this paper, we designed and implemented the first reinforcement learning framework for general WAF evasion tasks. The primary contributions of this work can be summarized as follows:

1. We proposed the first general WAF evasion framework with reinforcement learning approach.
2. We create an OpenAI Gym environment for WAF evasion tasks. This can facilitate us and other researchers to study and explore WAF evasion with reinforcement learning.
3. We conduct experiment with Proximal Policy Optimization (PPO) algorithm and successfully generate evasion payloads for both WAF-Brain (grey-box) and ModSecurity (black-box).

In the following sections, we first review existing works in Section 2, then we define this problem under reinforcement learning framework with Section 4. After presenting our environment and data in Section 5, we show our experiments and results in Section 6.

## 2 Related Work

### 2.1 Adversarial Machine Learning for Anti-Malware Evasion

Evasion of anti-malware detection is a very similar topic with our work and also a more explored area. Recent years, modern anti-malware detection algorithms take advantages of machine learning to construct either primary detection engines or as supplementary detection heuristics. However, such machine learning algorithms are vulnerable to intentional attacks. Several researches have proof that an attacker can bypass these machine learning detection algorithm by adversarial techniques. In 2017, Hu and Tan [8] introduced MalGAN to generate adversarial examples to attack black-box malware detection algorithms among the first attempt attacking portable executable malware models. MalGAN is a modified of GAN with a generator, a black-box detector and a substitute detector. The idea of the attack is letting the substitute model trained to reproduce outputs observed by probing certain inputs fed into the black-box detector. Then, the substitute model is used for gradient computation to produce evasive malware samples. Though this attack is reported 100% efficacy in bypassing the target detector, it has notable limitations, including requiring the complete knowledge of the feature space of the target malware detector and only focusing on LSTM variants. Later, Rosenberg et. al [10] proposed another end-to-end black-box method to generate adversarial examples to attack machine learning malware detectors, whose target is extended to multi-feature based malware detectors. The attack first creates a surrogate model using the target detector by Jacobian-based dataset augmentation, then generates adversarial examples using mimicry attack with white-box access to the surrogate model and using them against the attacked black-box model, by the transferability property.

### 2.2 Evading Black-Box Malware Detector with Reinforcement Learning

Apart from heuristic methods mentioned above, some recent works implement black-box attacks on malware detection algorithms using reinforce learning methods. Anderson et. al [1] pioneer using deep reinforcement learning to attack the malware detection engines in 2017. They design a series of actions to interact with the malware functions without prior knowledge of the structure, features and parameters of static PE malware detector. Their work also implemented the malware evasion environment as an extensible openAI Gym and performed deep Q-learning with agents utilizing a

Boltzmann exploration strategy. Later in 2019, Fang et. al [7] make further improvement on using deep Q-learning in evading anti-malware engines. Large dataset are used in training the agent, which is a complicated deep convolutional Q-network, and a 200% performance improvement is observed comparing to the Gym environment proposed in the pioneer work.

Even though our work shares similar idea with these works, the difference in targets (WAF vs. anti-malware) introduces totally different challenges. One key challenge is that web attack payloads are often short strings with strict grammar structures, which have much more limited manipulation space comparing to binary files.

### 2.3 WAF Evasion with Machine Learning

On the other hand, WAF evasion is a relatively underexplored topic. Earlier approaches include automata learning [3] and genetic programming based machine learning [2]. In the case of automata learning, the authors present SFADIFF, which is a black-box differential testing framework based on Symbolic Finite Automata (SFA) learning. This framework can be used to find differences between comparable programs, including but not restricting to WAFs and HTML/JavaScript parsing differences between several major browsers. The leveraged differences found by SFADIFF can lead to successful XSS attacks while evading WAFs. For the genetic programming based machine learning method, the authors typically focused on web application firewalls aiming at preventing SQLi attacks. Their main contribution is developing a SQLi grammer based on known SQLi attacks to date and an automated input generation technique. The details of the input generation technique include a context-free grammar for SQLi attacks, a random generation technique (RAN) which utilizes the syntax diagram generated from the context-free grammar and perform random selection at branching points after starting at the entry, and a machine learning based generation method that uses RAN to generate initial training data to learn a model predicting the likelihood with which tests can bypass the WAF. With recent trend of machine learning based WAF products, learning-based evasion study also appeared as Demetrio et. al [6] introduced WAF-A-MoLE, which is a tool for producing adversarial examples against WAFs by leveraging on a set of predefined syntactical mutations. The attack idea of WAF-A-MoLE is simple, it first starts with a failing test, that gets repeatedly transformed through the random application of mutation operators. Then the test will be modified, executed, compared and ordered. Finally, iterating among these steps until a successful test is found. They also produced a dataset of both sane and injection queries, which may be helpful to generating tests in later experiments using other methods evading WAFs.

## 3   Background Knowledge

In this section, we present necessary background knowledge about reinforce learning algorithm chosen for training the agent for evasion WAFs, which is the Proximal Policy Optimization Algorithm (PPO).

PPO is first introduced by Schulman et. al in 2017, aiming to retain the reliable performance of TRPO algorithms, while only using first-order optimization to reduce the computation complexity [12]. It is a model-free, on-policy, actor-critic policy-gradient method. For PPO algorithms, $\pi$ is the policy network which is optimized with regard to its parameter $\theta$. The policy network takes the state, $s$, as input and outputs an action. For discrete action spaces, the policy network will return actions drawn from samples of a probability distribution. When training the agent, actions are sampled randomly from the distribution for exploration and the finalized action is given by the mean of the actions during the training.

We then introduce the details of the PPO algorithms, including policy gradient method, actor-critic method applied in PPO. Policy gradient methods estimate the policy gradient and then using a gradient ascent algorithm to the gradient estimation. The gradients are estimated in a Monte Carlo (MC) scheme by running the policy in the environment to obtain samples of the policy loss $J(\theta)$ and

its gradient [15]:

$$J(\theta) = \mathbb{E}_{\tau \sim \pi_\theta(\tau)} \left[ \sum_t R(s_t, a_t) \right] = \mathbb{E}_{\tau \sim \pi_\theta(\tau)}[R(\tau)], \tag{1}$$

$$\nabla_\theta J(\theta) = \mathbb{E}_{\tau \sim \pi_\theta(\tau)} \left[ \left( \sum_{t=1}^{T} \nabla_\theta \log \pi_\theta(a_t|s_t) \right) R(\tau) \right] \tag{2}$$

where $\tau$ represents trajectories in the form $(s_1, a_1, s_2, a_2, \cdots, s_T, a_T)$. The gradients are then backpropagated to update $\theta$.

One central challenge for the policy gradient methods is to reduce the variance of the gradient estimation for making consistent progress in optimizing to better policy. The actor-critic method significantly impacts in this by reformulating the rewards in terms of advantage:

$$Q^\pi(s, a) = \sum_t \mathbb{E}_{\pi_\theta}[R(s_t, a_t)|s, a] \tag{3}$$

$$V^\pi(s) = \sum_t \mathbb{E}_{\pi_\theta}[R(s_t, a_t)|s] \tag{4}$$

$$A^\pi(s, a) = Q^\pi(s, a) - V^\pi(s) \tag{5}$$

The advantage function (5) measures the level of an action comparing to all other available actions in the state, for which good actions will get positive rewards and bad actions will get negative rewards. Thus, the estimation of the average reward is necessary and is done by the critic network, which is a separate neural network trained in a supervised way to predict the value function from the rewards in the gathered samples. Improvements like generalized advantage estimate (GAE) [11] are further employed to reduce variance of the advantage estimates. PPO also utilizes multi-actor scheme for sample gathering to increase the sample batch size.

PPO maximizes the surrogate objective function

$$L(\theta) = \hat{\mathbb{E}}_t \left[ \min \left( r_t(\theta)\hat{A}_t, clip(r_t(\theta), 1 - \epsilon, 1 + \epsilon)\hat{A}_t \right) \right], \tag{6}$$

where $\hat{A}$ and $\hat{\mathbb{E}}$ are the empirically obtained estimates of the advantage function and expectation, respectively, and $r_t(\theta)$ is the probability ratio defined as

$$r_t(\theta) = \frac{\pi_\theta(a_t, s_t)}{\pi_{\theta_{old}}(a_t, s_t)}. \tag{7}$$

---

**Algorithm 1** PPO, Actor-Critic Style

---
1: **for** iteration = $1, 2, \cdots$ **do**
2:     **for** actor = $1, 2, \cdots$ **do**
3:         Run policy $\pi_{\theta_{old}}$ in environment for $T$ timesteps
4:         Compute advantage estimates $\hat{A}_1, \cdots, \hat{A}_T$
5:     **end for**
6:     Optimize surrogate $L$ wrt. $\theta$
7:     $\theta_{old} \leftarrow \theta$
8: **end for**

---

As Vanilla policy gradients requires sample from optimized policy, which are not usable for the improved policy after one optimization step, PPO increases the sampling efficiency by using importance sampling method to obtain the expectation of samples gathered from an old policy $\pi_{\theta_{old}}$ under the refine-required new policy $\pi_\theta$. As the new policy is refined, the two policies will diverge and then increases the variance of the estimation, therefore, the old policy is periodically updated to match the new policy. This approach is valid if the state transition function is similar between two policies, and the requirement is ensured by clipping the probability ratio (7) between $[1 - \epsilon, 1 + \epsilon]$. This also gives a first-order approach to trust region optimization as the algorithm is not too greedy in choosing actions with positive advantages and not too quick to avoid actions with negative advantages from a small set of samples [5]. The minimum operator in (6) also ensures that the surrogate objective function remains a lower bound on the unclipped objective, and eliminates the increased preference for actions with negative advantage. A baseline PPO is stated in Algorithm 1.

# 4 Problem Definition

We formulate the WAF evasion problem into a reinforcement learning problem with Markov Decision Process (MDP). As an analogy to game playing, in the WAF evasion task, payload mutation is the game playing strategy, while the WAF represents the internal rule of the game, which tells whether we pass the game or not. Figure 1 illustrates the WAF evasion problem under the reinforcement learning framework. To be more specific, we define *state*, *action*, and *reward* as follows.

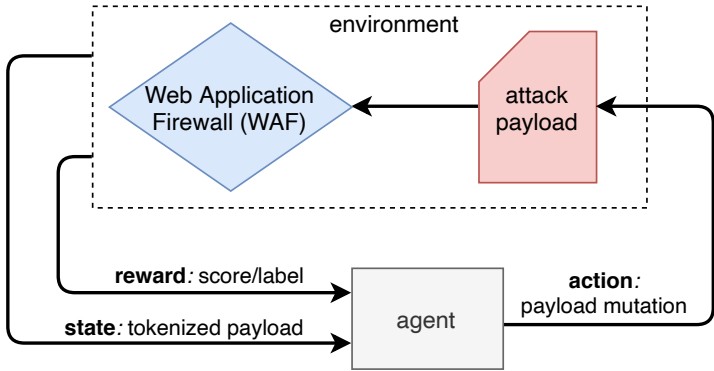

Figure 1: Formulation of the WAF evasion task as an reinforcement learning problem.

## 4.1 State

We consider the system that the agent learns to interact with any given payload. Further, we define the states that the agent observes to be feature vectors which comprises of three level of histograms, including token types, tokens, and characters. Figure 2 illustrate different levels of features with an example. Formally, let $\mathcal{T}$ denote a finite set of tokens. For a given original attack payload $P$, the set of possible program inputs $\mathcal{P}$ is then defined by a rule similar to the Kleene closure but with a finite set of tokens. We then define the states of our model by extracting the histograms of the token types, tokens, and characters (bytes) of an input attack payload as $\mathbf{H_k}$, $\mathbf{H_t}$ and $\mathbf{H_c}$ and concatenate the three histograms together to form the state $\mathcal{S} := [\mathbf{H_k} \ \mathbf{H_t} \ \mathbf{H_c}]^T$. In the following, $p \in \mathcal{P}$ denotes a mutated payload of a given original attack payload.

| Payload | | | | | | a' or 1 < 2 -- a | | | | | | |
|---|---|---|---|---|---|---|---|---|---|---|---|---|
| Characters | a | ' | ␣ | o | r | ␣ | 1 | < | 2 | ␣ | - - | | a |
| Tokens | STR | SP | OR | | SP | NUM | LT | NUM | SP | CMT | | |
| Token types | String | Space | Keyword | | Space | Comparison | | | Space | Comment | | |

Figure 2: Illustrative example of three levels of payload features.

## 4.2 Action

We define the set of possible actions $\mathcal{A}$ of our agent to be a discrete random variables mapping the payload of the input to probabilistic mutating rules

$$\mathcal{A} := \{a : \mathcal{P} \to (\mathcal{P} \times \mathcal{P}, \mathcal{F}, P) | a \sim \pi(p)\}, \tag{8}$$

where $\mathcal{F} = \sigma(\mathcal{P} \times \mathcal{P})$ denotes the $\sigma$-algebra of the sample space $(\mathcal{P} \times \mathcal{P})$ and P gives the probability for a given mutation rule. The mutation rule set is listed as follows.

- *space to comments*: randomly replace space with comments, or vice versa.
- *random case*: randomly swap cases for all letters in the payload.
- *swap keywords*: swap operators with its alternatives, e.g. swap "||" to " OR " or " || ".
- *swap integer base*: swap the base of the integer in the payload to its alternative, e.g. decimal to hexadecimal.

- *swap space to whitespace alternatives*: swap space with whitespace alternatives, e.g. `"\t"`, `"\n"`, `"\f"`, `"\v"`, `"\xa0"`.

- *rewrite comment*: rewrite the comments in the payload.

- *change tautologies*: rewrite a tautology to its alternate interpretation, e.g. `1 = 1` to `2 = 2`.

- *logical invariant*: adds an invariant Boolean condition to the payload, e.g. `something OR False`.

- *reset inline comments*: remove randomly chosen multi-line comment content.

### 4.3   Reward

We first set the maximum reward $R_{max} = 10$ and the minimum reward $R_{min} = 0$. Then by observing results given by the environment after the agent sending a mutated payload to the environment, we define rewards independently for two different environments in terms of their available returns:

1. Let $f_l \in \{0, 1\}$ be the classification function of a WAF that only return boolean labels, we define the reward to be $R_l = R_{max} \cdot f_l$. We have $R_l \in \{0, R_{max}\}$.

2. Let $f_s \in [0, 1]$ be the classification function of a WAF that returns a scalar score indicating the confidence level of a payload being malicious, and the score threshold for the WAF to classify a payload as malicious is $\theta \in (0, 1)$. We define the reward to be $R_s = \frac{\theta R_{max}}{\max\{f_s, \theta\}}$. We have $R_s \in [\theta R_{max}, R_{max}]$.

To encourage evasion with fewer mutation turns. We further update the reward over a mutation period using the reward function defined as $R = R_t - \sigma t$ where the value for $R_t$ is either given by $R_l$ or $R_s$ at step $t$ inside one episode, and $\sigma$ is the constant step penalty, which is set to be 0.1 in our environment.

## 5   Environment and Data

Our WAF evasion framework targets all kinds of WAFs in general. Techniques behind WAFs on the market fall under two categories: rule-based and ML-based. The former makes decision mainly based expert-defined rules while the latter use machine learning algorithm to classify benign and malicious data. Even though some of the WAFs have source code available, we always assume that adversaries have no access to the source code or the internal model of the target WAF. On the other hand, some WAFs return additional information like a scalar score apart from the Boolean label. We design our framework to be able to utilize the score information if available. Table 1 lists five mainstream WAFs with their categories and settings.

Table 1: Mainstream WAF Products with Various Categories and Settings

|  | ModSecurity[14] | WAF-Brain[4] | SQLiGoT[9] | Cloudflare WAF | Tencent WAF |
|---|---|---|---|---|---|
| Technique | Rule-based | ML-based | ML-based | Unknown | Unknown |
| Result type | Label | Score | Score | Label | Label |
| Open-source | Yes | Yes | Yes | No | No |
| Deployment | Local | Local | Local | Remote | Remote |

For experiments and evaluation purpose, we mainly consider open-source WAFs. The reason is that most commercial WAFs are cloud-based and can only be deployed remotely, while open-source WAFs can be deployed on our local machine, which brings much faster testing and training speed without network delay. After deploying the WAF locally, our agent program can directly interact with the WAF's programming interfaces, or send data to a simulated HTTP endpoint, which is processed by the WAF under testing. We then define rules to extract indicators of whether our requests are blocked by the WAF.

In this work, we select ModSecurity (rule-based) and WAF-Brain (ML-based) as our targets for experiments. These two are both open-source projects and are widely deployed or integrated in real world systems. ModSecurity uses token based signature for rule matching. WAF-Brain applies RNN classifier with raw character stream of the payload as the input. ModSecurity only returns a true or

false label while WAF-Brain also returns a confidence score as the additional information. We believe these two can represent two major categories of WAFs in the market.

To let the agent learn mutation strategies, we need to first provide some original attack vectors. We assume two types of environment settings. The first setting is that the agent will always learn to mutate a fixed payload. In our case is the most common SQL injection attack payload `1' or 1=1 -- a`. Each time the episode is over the environment will reset the payload to this fixed payload. The second setting is that the agent needs to learn the mutation against a pool of random payloads. We collect a list of payload from a public SQL injection dataset [13], which contains over 1000 payloads. Each time the episode is over the environment will randomly choice a payload from the list of payloads for reset.

With above mentioned targets and settings, we mainly use four different environments for experiments in this work. We refer to them as WAF-Brain (single payload), WAF-Brain (1k payloads), ModSecurity (single payload), ModSecurity (1k payloads)

## 6 Experiment and Result

Inspired by the gym-malware environment [1], we implement the WAF evasion environment as an extensible OpenAI Gym and trained it with Proximal Policy Optimization (PPO). Our framework shows its effectiveness of generating SQL attack payloads that can evade the detection of WAF-Brain and ModSecurity, one ML-based WAF with scores available and one rule-based WAF with only Boolean labels, both deployed locally.

### 6.1 Creating the Gym-WAF Environment

We create gym-waf, an OpenAI gym environment for general WAF evasion tasks. In this environment, we define uniform state representation, reward, and actions, which are general for different types of WAFs. For adopting this environment to a new WAF, the only customization required is to define an interface which tells the agent how to interact with the WAF. We already provided pre-defined abstract interface for remote/local WAFs with label/score result. Currently, we only implemented interface for WAF-Brain and ModSecurity, which are both local environments, one with score results and another with label results. We released gym-waf as an open-source project [2] to facilitate future research and exploration.

### 6.2 Training with PPO

We train the WAF evasion task against WAF-Brain and ModSecurity with two different environment settings. One with single fixed payload and one with random selection from a pool of 1000 payloads. The environment with a fixed payload is expected to be easier to train, while the latter environment setting is noisier and more challenging. Since ModSecurity only return labels, which gives very sparse rewards, its evasion is expected to be far more difficult reinforcement learning task. We set the learning rate to be $0.01$, mini-batch size to be $64$, entropy coefficient to be $10^{-4}$, clip range to be $0.3$. We run the training for $5 \times 10^5$ time steps on a machine with 20 CPU cores (2.4GHz). We then evaluate the successful evasion rate with 1000 payloads sampled from a pool of 10000 unseen SQL injection payloads. We compare the results with using an agent with random policy (random mutation) as the baseline. Our trained agent outperforms the baseline 17x (single payload) and 2x (1000 payloads) in the WAF-Brain evasion tasks. When it outperforms the baseline 44x in the ModSecurity evasion task with 1000 payloads, both agents fail to find any evasion in the single payload setting. This happens because the specific fixed payload `1' or 1=1 -- a` is a very typical SQL injection payload and is well blocked by ModSecurity's rules. It is possible that evasion does not exist for this particular payload. Table 2 shows the detail success rate of each environments.

### 6.3 Tuning of Hyperparameters

We conduct several experiments to find a set of suitable PPO hyperparameters. We use WAF-Brain with 1000 payloads as our targets for hyperparameter tuning. Since the WAF evasion task we defined in the gym-waf environment has discrete action space, we start with higher entropy coefficient to

---

[2]https://github.com/sanebow/gym-waf

Table 2: Success Rate of Evasion Tasks (PPO v.s. Random Mutation)

|  | WAF-Brain (single payload) | WAF-Brain (1k payloads) | ModSecurity (single payload) | ModSecurity (1k payloads) |
|---|---|---|---|---|
| Random agent | 2% | 10% | 0% | 0.2% |
| PPO agent | 35% | 20% | 0% | 8.7% |

prevent single action dominating the policy and set smaller minibatch size. Then we run the training with a range of different hyperparameter values, each for 500,000 time steps. Figure 3 shows the training progress with different settings. Based on these experiments, we find that low learning rate in our task can result in less stable training process, so we choose the learning rate to be $10^{-2}$. Minibatch size does not have significant impact on the performance, but larger size appears to fluctuate the training. We pick an intermediate minibatch size 64 for marginal performance improvements. For entropy coefficient, we find that setting it to $10^{-4}$ yields a slightly faster learning process. Lower clip range results in slower training, and setting it to 0.3 yields reasonable performance.

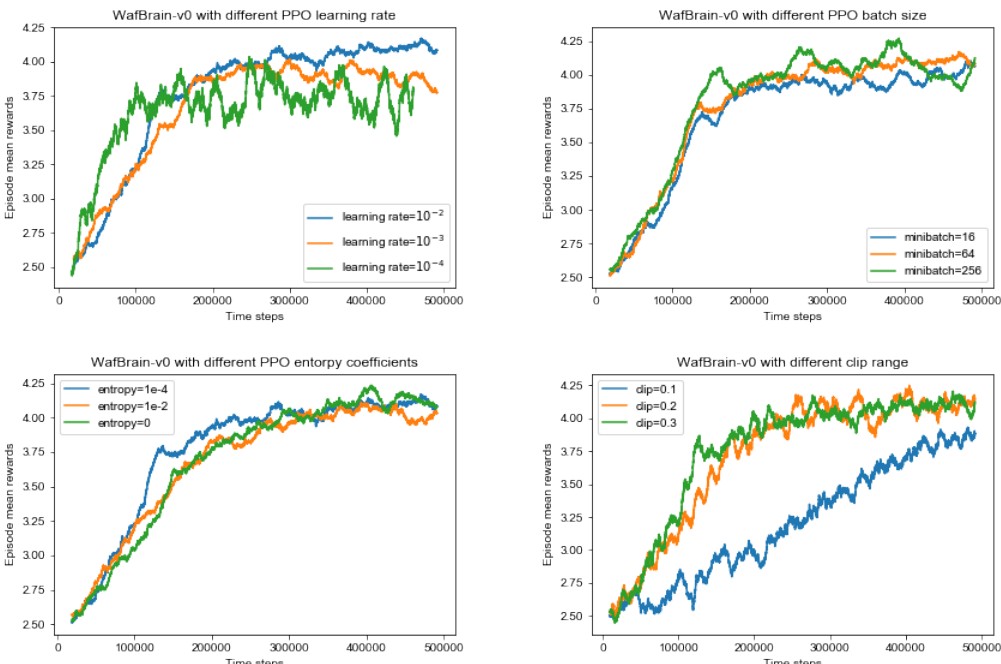

Figure 3: Training curve with different hyperparameters.

## 6.4 Comparison: DQN vs PPO

We also conduct some experiments with Deep Q-Network (DQN), which is an off-policy learning algorithm. In this section we present performance comparison between DQN and PPO on the WAF-Brain evasion task.

We set up the DQN with three hidden layers of size 1024, 256 and 32. We also add a dropout layer after input to make the model less sensitive. In our training experiment, max episode length is set to be 20 and batch size is 16. For PPO, we use the hyperparameters stated in Section 6.3. We trained both algorithm on the WAF-Brain evasion task for 100 minutes. PPO performs significantly better as shown in Figure 4.

## 6.5 Analyzing the Discovered Evasion Payloads

Our trained agent generated thousands of evasion payloads and we witnessed various evasion strategies it applied. Some of them are quite surprising and inspiring. Table 3 shows a sample of the generated

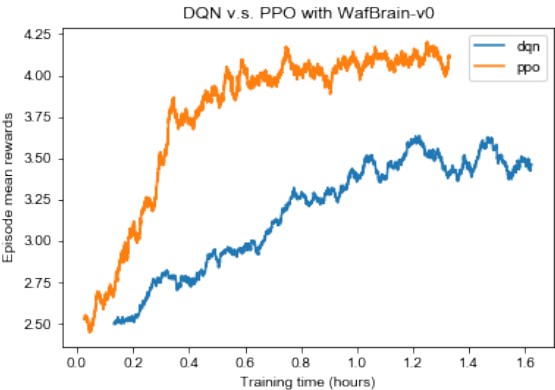

Figure 4: DQN vs PPO on training WAF-Brain with 1000 payloads.

evasion payloads. By looking at the generated payloads and analyzing the techniques behind, we identified some interesting phenomenon and spotted several weaknesses in existing WAF products.

- The successful evasion payloads for WAF-Brain contains no obvious common pattern. They often combine several different mutation strategies.
- The trained WAF-Brain evasion agent has the preference to insert invisible characters like \x0B and \xA0 into the payload. It turns out there is a bug in WAF-Brain's code that leads to classification failure when encountering certain invisible characters.
- Trained ModSecurity models stably replace the logical expressions like "1=1" with logical invariant in the form of "x NOT = y". With some investigation, we found that it is a specialized syntax for T-SQL and ModSecurity missed it in its rule set.

Different techniques used in two WAFs can actually explain our first finding. More specifically, ModSecurity relies on hard-coded rule sets while WAF-Brain applies machine learning classifier. Thus, combining various mutations are more promising way of evasion. On the other hand, a corner case missed by the rules of rule-based WAF can be the key for its evasion. The operation "NOT =" is a rarely used operator in T-SQL and is not included in the rule sets of ModSecurity. Besides, silly bugs in WAF's code may give backdoors to attackers, and we show that our reinforce learning approach can potentially help human discover bugs in WAFs more effectively.

Table 3: Examples of Evasion Payloads Discovered by PPO Model

| Evasion payload | Target WAF | Episode lengh |
|---|---|---|
| 1 or/*5x$!*/6891=6891/*5x$!*/ AND "$" NOT = "$c"– a | ModSecurity | 12 |
| 1 or 1=1 AND "$" NOT = "$c"– a | ModSecurity | 4 |
| 1/*5X$!*/OR/**/1=1 – A | WAF-Brain | 5 |
| 1/*5X$!*/OR 6891=6891 aND "$" nOt LiKe "$C"– | WAF-Brain | 8 |

## 7 Future Work and Conclusion

In this work, we explored the possibility of solving the WAF evasion task with reinforcement learning. We designed and implemented an extensive WAF evasion environment and applied fine-tuned PPO to train it. Our trained agent can significantly outperform random mutation baseline for both WAF-Brain and ModSecurity. The results even display previous-unknown evasion strategies that can help identify weakness in existing WAF products. One major direction for improvement is to design grammar-aware state representation and mutation. Current token-histogram based state representation may not be able to estimate the state accurately, and current predefined mutation strategies limit the action space. Another direction that worth more exploration is the improvement of training algorithm in no-score environments like ModSecurity, where rewards are extremely sparse. Finally, we plan to

evaluate our evasion framework against more WAFs, including those cloud-based WAFs with only remote access.

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
