# OpenReview forum: "Evading Web Application Firewalls with Reinforcement Learning"
_CUHK.edu.hk/2021/Course/IERG5350_

### Official Review · AnonReviewer1 · 2020-12-16
**An outstanding attempt to perform WAF evasion tasks with reinforcement learning**

**Rating:** 9
**Confidence:** 5

**Review:**

__General__

The project analyzed WAF evasion with reinforcement learning which is an underexplored field. The problem is clearly defined and successfully solved with both PPO and DQN. Evaluations of both methods are provided including comprehensive hyperparameter tuning and algorithm-wide performance comparison. As contributions, the project provides a gym environment for WAF evasion tasks and a validated reinforcement learning framework for WAF evasion.

__Significance__

The project pioneers a brand new field that has the potentials for further in-depth researches. It is a successful integration of reinforcement learning and security while the environment and agents are reusable. The future targets concerning syntax awareness, no-score environments, and versatility are very clear.

__Flaw__

I can't think of any other than minor grammar and wording mistakes. (e.g. advisory should be adversary?)

---

### Official Review · AnonReviewer2 · 2020-12-16
**Evading Web Application Firewalls with Reinforcement Learning**

**Rating:** 9
**Confidence:** 3

**Review:**

Summary: This paper tries to use reinforcement learning to automate the process of WAF evasion. A reinforcement learning environment based on the gym is created. Two methods (PPO and DQN) are utilized in this environment and a comparison between these two methods is also shown.
Technical quality: The definition of the problem is clear and two methods are used.
Novelty: The authors create a gym environment for WAF evasion tasks and propose a framework with a reinforcement learning approach.

comments:
1. provide a more detailed analysis of the comparison of DQN and PPO

---

### Official Review · AnonReviewer3 · 2020-12-20
**A new enviroment for Web Application Firewalls evasion task to apply RL algorithm**

**Rating:** 8
**Confidence:** 4

**Review:**

**General:**

This paper presents a new enviroment for Web Application Firewalls evasion task. Previous work shares similar idea, but focuses on anti-malware while this paper focuses on WAF evasion. The authors elaborate the design of their environment, and conduct ablation study of hyperparameters. They train the agent with PPO on two open-source datasets and show the effectiveness of their agents to successfully predicate evasion payload. Lastly, they analyze the action of the trained agents.


**Quality:**

This paper is well organized and provides sufficient experiments to show the effectiveness to apply RL in this environment.

**Clarity:**

The paper is clearly written. I can understand it even without any background in security.

**Originality:**

It shares similar idea with previous work, but focuses on different task. It meets the requirement for a course project.

**Significance:**

The environment is helpful for those who want to discover evasion payload without labour and intelligence.

**Pros:**
1. A totally new environment for WAF task.
2. Sufficient experiments and ablation study.
3. The analysis of trained agent is quite interesting and shows the effectivenss.

**Cons:**
1. There are some typos and ambiguous descriptions, please find in the comments.

**Comments:**
1. In P5. Section 4.1. What is the character of payload? Please give a example.
2. In P6. Section 4.3. $R_l \in \set{0, 1}$ should be $R_l \in \set{0, R_{max}}$.
3. In P6. Section 4.3. What's the confidence for, malicious or not? Please describe it more clearly.
4. In P6. Section 4.3. $R_s \in [0, 1]$ should be $R_s \in \[\theta R_{max}, R_{max}\]$ according to the definition.
5. In P8. Section 6.3. "we choice" should be "we choose".
6. In P8. Section 6.3. $10^-4$ should be $10^{-4}$.